# Silver Nanoshells with Optimized Infrared Optical Response: Synthesis for Thin-Shell Formation, and Optical/Thermal Properties after Embedding in Polymeric Films

**DOI:** 10.3390/nano13030614

**Published:** 2023-02-03

**Authors:** Laurent Lermusiaux, Lucien Roach, Moncef Lehtihet, Marie Plissonneau, Laure Bertry, Valérie Buissette, Thierry Le Mercier, Etienne Duguet, Glenna L. Drisko, Jacques Leng, Mona Tréguer-Delapierre

**Affiliations:** 1University Bordeaux, CNRS, Bordeaux INP, ICMCB, UMR 5026, 33600 Pessac, France; 2University Bordeaux, CNRS, Solvay, LOF, UMR 5258, 33608 Pessac, France; 3Solvay R&I, 52 rue de la Haie Coq, 93306 Aubervilliers, France

**Keywords:** nanoshells, silver, ultrathin, infrared, synthesis, thin film, solar energy control

## Abstract

We describe a new approach to making ultrathin Ag nanoshells with a higher level of extinction in the infrared than in the visible. The combination of near-infrared active ultrathin nanoshells with their isotropic optical properties is of interest for energy-saving applications. For such applications, the morphology must be precisely controlled, since the optical response is sensitive to nanometer-scale variations. To achieve this precision, we use a multi-step, reproducible, colloidal chemical synthesis. It includes the reduction of Tollens’ reactant onto Sn^2+^-sensitized silica particles, followed by silver-nitrate reduction by formaldehyde and ammonia. The smooth shells are about 10 nm thick, on average, and have different morphologies: continuous, percolated, and patchy, depending on the quantity of the silver nitrate used. The shell-formation mechanism, studied by optical spectroscopy and high-resolution microscopy, seems to consist of two steps: the formation of very thin and flat patches, followed by their guided regrowth around the silica particle, which is favored by a high reaction rate. The optical and thermal properties of the core-shell particles, embedded in a transparent poly(vinylpyrrolidone) film on a glass substrate, were also investigated. We found that the Ag-nanoshell films can convert 30% of the power of incident near-infrared light into heat, making them very suitable in window glazing for radiative screening from solar light.

## 1. Introduction

Metallic nanoshells, which are generally gold or silver, possess tunable optical properties controlled primarily by the diameter of the dielectric core and the thickness of the metallic shell [1]. In general, an increased core diameter or a thinner shell red-shifts and broadens the plasmon resonance, moving it into the near-infrared (NIR) and infrared (IR) regions [2,3], making these systems attractive particles for therapeutic applications [4,5]. In comparison to gold, silver exhibits a sharper and stronger plasmon resonance [6] and a 10-to-10^3^-fold greater surface-enhanced Raman scattering (SERS) enhancement for isolated particles [7], while being less expensive. Silver is therefore an attractive metal for industrial applications, such as plasmonic glasses and films [8]. More synthetic approaches have been proposed to prepare nanoshells made of silver than of gold [1], resulting in various morphologies of silver shells, such as smooth [6,9,10], bumpy [11,12,13], spiky [6,10], particulate [14,15,16,17,18], and ultrathin [19]. This structural diversity results in an even larger range of optical properties [12], which have been used in applications such as photocatalysis [16], photonic materials [20], bio-imaging [21], and drug detection. [15] Moreover, silver nanoshells, especially those with bumpy or spiky surfaces, also possess a strong electromagnetic field enhancement, which is used for many SERS [9,12,22] or SERS-based applications, such as cell tracking in live animals [11] and the label-free detection of pesticides [13]. Another potentially interesting and largely unexplored application is the radiative screening of infrared light.

Over the broad range of reported silver-nanoshell geometries, only a few exhibit stronger extinction in the NIR than in the visible [9,19]. They usually possess a maximum intensity peak in the visible, especially if the core is small [21] or if the shell is discontinuous [23]. but most often, they present a broadband extinction over both the visible and the IR [11,13,24]; some studies show similar spectra but omit the IR extinction [24,25]. Overall, the most direct way to obtain a stronger absorption in the IR than in the visible is to synthesize an ultrathin silver nanoshell over a relatively large silica core (above 150 nm) [19]. Furthermore, some gold and silver nanoparticles have strong absorption in the infrared, such as gold bipyramids [26] or silver plates [27]; however, the isotropic optical properties of nanoshells make them more attractive for some applications, including the fabrication of plasmonic surfaces and films [8].

The synthesis of thin, continuous Ag or Au shells around a core is difficult because of the poor wettability of these metals on SiO_2_ surfaces. The shell formation typically proceeds through an island-growth mechanism, leading to inhomogeneities in surface coverage, as nucleation sites tend to grow isotropically rather than as a continuous layer on the SiO_2_ surface. The most common approach to synthesizing a silver shell is to reduce silver onto gold-seeded silica particles [6,19,21,28,29,30]. This approach has been used to produce an ultrathin shell (9.8 nm thick) thanks to the addition of a polymer additive (e.g., polyethylenimine) during the shell formation [19], which plays three different roles: facilitating shell growth, regulating the reaction rates, and stabilizing the particles. In practice, the polymer is used to slow the shell-growth reaction so that it occurs uniformly. However, this approach is limited by the continued presence of gold seeds within the shell, which detrimentally influences the optical properties of the nanoshell due to the interband transitions of Au [1,22], especially if it is very thin. Therefore, two main approaches have been developed to synthesize nanoshells made only of silver: direct- or seeded-growth synthesis.

For direct synthesis, several one- or two-step approaches were developed using different physicochemical mechanisms, such as electroless plating [20,22,28], a template-activated strategy [31], methods using the Tollens’ process [32], ultrasonic electrodeposition [24], layer-by-layer deposition [14,23], the amine-assisted reduction of silver onto thiol-functionalized silica particles [11,12,13], and a ‘rapid chemical method’ using neither coupling agents nor a functionalized core [10]. The short reaction time of these approaches makes them very convenient; however, the fabricated shells are usually thicker than 20 nm.

To produce thin homogenous shells using a seeded-growth approach requires a high density of seeds on the dielectric particles. However, depositing silver seeds onto functionalized silica particles does not produce sufficiently high seed densities. One reason for this is that Ag seeds have low chemical stability, as they oxidize rapidly. The regrowth of sparse silver-seed deposits results in large shells [23,33,34]. Several routes were developed to obtain a high silver particle density on silica particles. These include using glucose to reduce silver at high pH values [35], or directly onto the thiol-functionalized particles [36], sonochemical deposition [37], and pretreatment steps in electroless plating [38]. The latter method uses the reduction of ammoniacal silver onto Sn^2+^-sensitized particles and was used to synthesize Ag nanoshells [9,39,40]. The regrowth of the seeds can be rapid (by the reduction of silver nitrate by formaldehyde and ammonia), which produces smooth shells as thin as 20 nm [9,39], or slow (using ammoniacal silver nitrate and formaldehyde in ethanol), producing slightly bumpy surfaces as thin as 14 nm [40]. Finally, the UV-Vis-NIR-extinction spectra of the smooth 20-nanometer-thick silver nanoshells show higher levels of extinction in the IR than in the visible, with a peak at 800 nm, making this synthetic approach the most promising for obtaining ultrathin shells with a high level of IR extinction.

In this paper, we present a protocol to deposit ultrathin silver nanoshells onto silica particles, which are either continuous or composed of large and thin patches. We used a seeded approach to first form a packed layer of silver seeds on the silica surface, which are regrown into a silver shell. We then combined optical spectroscopy and high-resolution electron microscopy to study the unusual synthesis mechanism of the particles during shell formation. We show the effects of different parameters on the synthesis products to provide an optimized protocol. Finally, we investigated the optical and thermal properties of the particles embedded within a continuous, transparent polymeric matrix and examined their potential to limit infrared transmission and prevent heating by incident sunlight.

## 2. Materials and Methods

### 2.1. Materials

Most of the reagents were purchased from Sigma-Aldrich and used as received. These included l-arginine (98.5%), tetraethoxysilane (TEOS, 99%), ammonium hydroxide (NH_4_OH, 28–30 wt. % in water), sodium hydroxide (NaOH, 98%), Tin(II) chloride dehydrate (SnCl_2_⋅2H_2_O, 98%), poly(vinylpyrrolidone) (PVP, 44,000 g⋅mol^−1^, 98%), silver nitrate (AgNO_3_, ≥99.0%), hydrochloric acid (HCl, 37%), and formaldehyde (HCHO, 37 wt % in water, containing 10–15 wt. % methanol as a stabilizer). The PVP (360,000 g⋅mol^−1^, 98%) used for the thin-film preparation was purchased from Merck. Aqueous solutions were prepared using deionized water, prepared using a Type 1 Milli-Q water-purification system (18.2 MΩ⋅cm).

### 2.2. Synthesis

Silica particles were synthesized by following a previously published protocol [41].

#### 2.2.1. Synthesis of the Silica Seeds

The l-arginine aqueous solution (100 mL, 6 mM) was placed in a 150-milliliter double-walled vial equipped with a reflux condenser. When the temperature stabilized at 60 °C, TEOS (10 mL) was added. The heating and stirring were stopped when the TEOS upper phase disappeared.

#### 2.2.2. Regrowth of the Silica Seeds

In total, 455 mL of ethanol (99%), 33 mL of ammonia (28%), and 10 mL of an aqueous dispersion of seeds were mixed and placed under stirring. The TEOS (20 mL) was added to the dispersion with a syringe pump at a rate of 0.5 mL⋅h^−1^. Once finished, another 20 mL of TEOS was added using the same protocol. The synthesized seeds were approximately 125 nm in diameter. The dispersion was centrifuged twice (10 min at 9000× *g*) and the particles were redispersed in 55 mL of ethanol (silica-particle dispersion).

#### 2.2.3. Additional Regrowth of the Silica Particles

Ethanol (450 mL, 99%), ammonia (34 mL, 28%), water (10 mL), and the regrown silica-particle dispersion (5 mL) were added and stirred. The TEOS (20 mL) was added to the dispersion with a syringe pump at a rate of 0.5 mL⋅h^−1^. The dispersion was centrifuged three times and the particles were redispersed in ethanol each time. The final volume of the silica-nanoparticle dispersion was 150 mL and the silica concentration was measured at 50.75 mg⋅mL^−1^. The synthesized particles were approximately 200 nm in diameter.

#### 2.2.4. Sn-Sensitization of Silica Particles

The silica-particle dispersion (2 mL, corresponding to 101.5 mg of particles) was dispersed in a 10-milliliter solution containing 53 mM Sn^2+^ and 10 mM HCl, after which it was agitated for 30 min. Next, the solution was centrifuged three times and the particles were redispersed in 2 mL of water.

#### 2.2.5. Preparation of Tollens’ Reagent

Tollens’ reagent (270 mM) was prepared by following a previously published protocol [42]. To 10.1 mL of water, 5.4 mL of 1 M AgNO_3_, 2.2 mL of 3 M NaOH, and 2.2 mL of ammonia (28%) were added, under stirring. The mixture first turned a murky brown, after which it returned to its previous colorless and transparent state over a few minutes. The mixture was used on the day of preparation.

#### 2.2.6. Synthesis of Ag Seeds on Sn-Functionalized Silica Particles

The Sn-sensitized particles (1.5 mL) were added to 20 mL of the Tollens’ reactant, under vigorous stirring. The dispersion immediately turned black. Stirring was maintained for 2 min and, subsequently, the dispersion was centrifuged three times, and the particles were redispersed in 10 mL of water. Particles can be used directly or within a week of preparation. To avoid problems arising from the detachment of seeds during this period, the dispersion was centrifuged and re-dispersed in water immediately before performing the last step.

#### 2.2.7. Formation of the Thin Silver Shell

In total, 1 mL PVP (40,000 g⋅mol^−1^, 10 g⋅L^−1^), 75 µL of Ag-seed-functionalized silica particles, between 10 and 50 µL of AgNO_3_ (100 mM), 50 µL formaldehyde (37% in water), and 20 µL ammonia (28%) were added successively to a volume of water (calculated to obtain a final volume of 10 mL) under vigorous stirring. The dispersion changed color within 1 min. It was then centrifuged twice and the particles were redispersed in 10 mL of PVP solution (40,000 g⋅mol^−1^, 10 g⋅L^−1^).

### 2.3. Composite-Film Preparation

Thin films made of the core-shell particles embedded in a PVP polymeric matrix were prepared by blade-coating in an evaporative regime (i.e., sufficiently slow for evaporation to solidify the film as it was deposited on the glass substrate). Two borosilicate glass plates with dimensions of 2 × 3 inches were placed close together. The first (the blade) was placed 3° from horizontal, 300 μm above the second plate (the substrate), which was placed horizontally on a linear stage, allowing substrate displacement relative to the blade. The linear stage was fitted with a heater to control the substrate temperature (fixed at 60 °C). The nanoshell dispersion was then redispersed into an aqueous solution of PVP (*M*_w_ = 360,000 g⋅mol^−1^). This dispersion was placed between the blade and the substrate to form a reservoir of liquid. The substrate was then moved at 1 μm⋅s^−1^ relative to the blade. The cast liquid yielded uniform thin films of about *L =* (20 ± 3) μm in thickness. The thickness was measured by scanning electron microscopy. Two different films with increasing amounts of particles *ρ*_1_ × *L* = 3.3 μm^−2^ and *ρ*_2_ × *L* = 10 μm^−2^ (*ρ_i_* = particle density per μm^3^) were prepared using 79 and 240 μL of Ag nanoshells (3.1 × 10^11^ mL^−1^), respectively, in 1 mL of PVP solution (5 wt%). The core-shell particles used to fabricate the films were produced using a volume of 28 μL of AgNO_3_ (100 mM) during the formation of the thin silver shell.

### 2.4. Characterization

Scanning transmission electron microscopy (STEM) studies were performed using a cold-FEG JEM-ARM200F (JEOL, Tokyo, Japan) operated at 200 kV equipped with a probe Cs corrector reaching a spatial resolution of 0.078 nm. Energy dispersive spectroscopy (EDX) spectra were recorded on a CENTURIO SDD detector (JEOL, Tokyo, Japan). We prepared the samples by depositing one drop (∼4 μL) of the colloidal dispersion onto a conventional carbon-coated copper grid and then air-drying the grids at room temperature. Prepared grids were stored in a closed box to prevent dust accumulation. Transmission electron microscopy (TEM) experiments were performed using a JEM 1400+ (JEOL, Tokyo, Japan) at 120 kV using a LaB_6_ filament. High-resolution (HR) TEM experiments were performed using a JEM 2200FS FEG HR 200 kV (JEOL, Tokyo, Japan). Extinction spectra of the nanoshell dispersions were recorded using a Shimadzu UV-3600 UV-Vis-NIR spectrometer. The extinction spectra of the thin films were measured using a microspectrophotometer, SV-5200 (JASCO Inc., OK, USA). Its 30-watt deuterium lamp and 20-watt halogen lamp allow spectral acquisition over a wavelength range of 200 nm to 2700 nm.

Thermal measurements of the thin films were recorded within an enclosure, in which the samples could be illuminated with a solar light source, as depicted in Figure 1. A HAL-320 solar source (Asahi Spectra, CA, USA) was used to illuminate the composite films on a glass substrate, and its beam was directed at the upper panel of the enclosure using a waveguide and a collimating lens. A vertical bar with a mobile blade holder was used to place the films at a fixed distance of 27 cm from the lens. This distance, calibrated by the supplier of the solar source, corresponded to a distance at which the incident beam possessed an intensity *I_0_* = 1000 W⋅m^−2^ (as the beam was divergent). The enclosure ensured that no external radiation could illuminate the thin film. As the incoming light illuminated the sample, the thin film absorbed a section of the solar spectrum, and the glass substrate absorbed some of the UV light, resulting in a temperature increase in both the film and the glass substrate. The system was allowed to stabilize until a steady state was reached. The temperature was measured above the film and below the glass slide (we only report the latter in the following sections). A K-type thermocouple was fixed to the rear face of the glass substrate, and temperature changes were acquired using a 24-bit National Instruments module (National Instruments, TX, USA) using LabView software. The thin-film temperature was more complicated to measure as the tip of the thermocouple was greater than the thicknesses of the thin films, leading to uncertainty in the measurements. However, as the thickness of the glass substrate was approximately 1 mm, its temperature could still be accurately measured, providing useful information on the heat generation within the thin film. Additionally, a high-pass optical filter was used after the collimating lens to block UV and visible light, letting only the NIR part of the solar spectrum reach the thin film. For the most advanced characterization, a Peltier module was used to collect the power transmitted across the film on glass, along with the thermal emission.

## 3. Results

### 3.1. Silver-Seeded Silica Particles

The approach developed to synthesize ultrathin silver nanoshells consists of an electroless plating process and was performed in several steps (Figure 2): (1) the sensitization of the silica particles with Sn^2+^; (2) the reduction of a high density of silver seeds directly onto the particles using the Tollens’ reagent; and (3) the reduction of silver nitrate by the successive addition of formaldehyde and ammonia. Although there are several steps, the entire shell synthesis can easily be performed within one day.

First, functionalization of 200-nanometer silica particles with Sn^2+^ was performed by simply mixing SnCl_2_ and HCl with freshly synthesized particles [40]. Next, the silica particles were directly added to a highly concentrated Tollens’ solution, resulting in an immediate color change from white to a very dark yellow, with the particles rapidly sedimenting. The surface of the silica particles had a very dense coating of silver seeds (Figure 1c,d), with Sn still detectable on the surface (Figure 1b). The UV–Vis–NIR spectroscopy of the particles showed a peak centered at 390 nm, arising from the Ag seeds, which added to the silica-particle signal (Figure 1a). We observed that with time, some seeds detached from the surface of the silica. It was crucial to centrifuge the silver-seeded silica particles to remove any unattached seeds just before performing the growth step. Otherwise, the following silver reduction also occurred on the free seeds, resulting in large silver particles as by-products. However, the initial seed density was so high that seed detachment did not seem to affect the quality of the subsequent nanoshell synthesis.

### 3.2. Ultrathin Silver Nanoshell

Ultrathin shells were produced in one step by successively adding silver nitrate, formaldehyde, and ammonia to a diluted dispersion of the silver-seeded silica particles. The color changed from light yellow to blue in seconds (see inset in Figure 2a). The UV–Vis–NIR spectroscopy showed a high level of extinction in the NIR region (between 800 and 1350 nm in Figure 2a). The extinction was approximately three times lower in the visible and increased slowly from 520 nm to 800 nm. The high-resolution electron microscopy of the particles showed that the silica particles were covered with an ultrathin, polycrystalline, and smooth silver shell (Figure 2). Although most of the particles had a complete shell, we also observed other shell morphologies such as particles with percolated, or discontinuous shells (Figure 2b) and others covered by a few ultrathin-isolated patches (Figure 2c). We also observed a few silver particles as by-products arising from the regrowth of the detached seeds in the dispersion (Figure 2d). The thickness of the nanoshells was approximately 10 nm on average, meaning there were many regions in which the shell was thinner, making it semi-transparent in the TEM images (Figure 2c–f).

### 3.3. Synthesis Mechanism

The different shell morphologies obtained using a seed-mediated approach are quite surprising, as they are very different from those of thin [43] or ultrathin [44,45] gold shells made from gold-seeded silica particles. In solution-based mechanisms, the regrowth of the seeds results in the formation of isolated gold islands that later merge into a shell (Au scheme) [43]. Alternatively, the homogenous growth of the seeds from a gaseous reducing agent forms continuous, yet slightly bumpy, nanoshells [44]. Here, the very smooth metallic surfaces obtained over a large area suggest another synthesis mechanism, which was studied by producing particles using different amounts of silver nitrate to regrow the shell. The optical properties of these particles were studied by UV–Vis–NIR spectroscopy (Figure 3). For the smallest volume of silver -nitrate solution used (four times less than that of the particles shown in Figure 2), we observed a wide peak centered at 1100 nm. With increasing quantities of silver, the resonance progressively blueshifted, and increased in intensity. Additionally, a dip was observed in the UV region due to the 4d–5s interband transition for Ag, alongside a weak plasmonic peak, which has been categorized as a quadrupolar mode by other authors [46].

Up to a certain volume of silver, the main extinction was mostly located in the NIR before shifting into the visible for a 30-nanometer-thick shell (40 μL, Figure 3). This behavior was completely different from that observed with the gold nanoshells. In these, the shell formation first exhibited a red shift in the visible resonance [43]. A blue shift in the resonance took place later, when the continuous metallic shell was formed. The optical results obtained here match numerical simulations that describe a continuous redshift in the extinction with decreasing shell thickness [2,3]. However, they do not originate from the same phenomenon as that in our experiments. In fact, for low quantities of silver, the TEM images showed several independent ultrathin patches of silver that partially covered the particle surfaces (Figure 4). The rest of the surface still contained silver seeds. The optical properties of the individual patches are likely to be comparable to those observed in ultrathin anisotropic silver nanoparticles, but the overall optical response of the nanoshell is strongly affected by plasmonic coupling between the patches [27]. For larger quantities of silver, the size of the patches increases drastically, whereas the thickness is relatively unchanged. The shell thickness starts to increase only when most of the surface is covered by silver. Because the reaction occurs under vigorous stirring, and takes place nearly instantaneously, we assume that this mechanism of producing a thin shell is supported by a high reaction rate. Conversely, slowing the reaction rate enables the synthesis of an ultrathin silver shell onto gold-seeded silica shells [19], demonstrating the presence of distinct mechanisms. Overall, this synthesis mechanism seems to proceed via the following steps: first, a rapid reduction of silver onto the seeds, forming initial patches on the silica surface, and, secondly, the occurrence of silver reduction at the edge of the patches, parallel to the silica surface (Ag scheme, Figure 4).

### 3.4. The Effects of Various Synthetic Parameters

The regrowth step is very sensitive to the following three parameters: the quantity of ammonia, the quantity of formaldehyde, and the time between the formaldehyde and ammonia injections. We varied each of these parameters independently and studied the resulting changes in the optical properties of the nanoshells. First, for a given amount of formaldehyde, there was an optimal ammonia volume that produced the desired structure (Figure 5a). The range of ammonia concentrations producing good nanoshells was quite narrow. By either doubling or dividing by four the optimized volume, the samples exhibited decreased extinction in the IR compared to the visible. Second, we found that increased quantities of formaldehyde resulted in better optical properties with an increased extinction, especially in the NIR (Figure 5b). Other experiments showed that excessive formaldehyde is also detrimental to the synthesis (data not shown). All the formaldehyde volumes used in these experiments corresponded to a very large excess compared to the Ag salt. A higher formaldehyde concentration may be one way to increase the shell-growth-reaction rate, favoring ultrathin shells.

The last key parameter is the time delay between the injections of formaldehyde and ammonia, which must be minimized (Figure 5c) [39]. As the time between the addition of these two products increased, we observed a blueshift in the extinction peak and a decrease in the extinction in the IR range. Thus, varying the time delay between injections allows the optical properties of the nanostructures to be easily tuned. Finally, there are a few other parameters that affect the final nanoshell morphology. Firstly, we observed that it is possible to add the ammonia before the formaldehyde, if the time delay is kept short. However, pre-mixing formaldehyde and ammonia does not produce good results, probably because they react together [47]. Secondly, the regrowth reaction was performed at room temperature, but a slightly increased temperature could also increase the reaction rate and favor the formation of the thin shell. Thirdly, the concentration of silver-seeded silica particles was found to be important. Finally, as previously mentioned, seed detachment from the particle surface was observed over time, and although little effect was observed on the optical properties of the nanoshell, a more systematic study of the effects of the seed density on the shell morphology would provide useful information on the synthesis of ultrathin nanoshells.

### 3.5. Optical and Thermal Properties of Dispersed Ag Nanoshells in Polymer Films on Glass

The nanoshells were dispersed in a thin polymer film deposited on a glass substrate by blade-coating. The optical properties of films doped with increasing amounts of particles were investigated. Figure 6 presents images of the films containing different particle densities (*ρ*_0_ = 0 μm^−2^, *ρ*_1_ = 3.3 μm^−2^, *ρ*_2_ = 10 μm^−2^), alongside their optical spectra. The reference sample (*ρ*_0_) exhibited high transparency in the visible range with a slight reduction in the NIR (not shown). By comparison, the films containing dispersions of the Ag nanoshells in the polymeric matrix exhibited a much greater difference in transmission between the visible and the NIR, efficiently reducing NIR transmittance. The attenuation band was wider and redshifted in comparison to the extinction of the nanoshells in solution. This shift was attributed to the change in the refractive index of the surrounding medium. Not surprisingly, the attenuation increased with increasing particle concentrations.

The thermal properties of the films were also measured (Table 1). Interestingly, it was found that, for the polymeric film with the highest concentration of particles (*ρ*_2_), a significant amount of solar power was absorbed and converted into heat. The solar-heat-gain coefficient (the ratio of the transmitted power to the incident illumination power, Table 1) demonstrated that up to 40% of the illumination power was absorbed at the highest nanoshell density, *ρ*_2_. A high-pass filter was used to illuminate the sample with only NIR light (>750 nm). In this spectral region, 30% of the NIR light was absorbed and transformed into heat, diminishing the amount of non-visible light transmitted across the composite film on the glass substrate. Hence, polymeric films doped with Ag nanoshells offer a promising level of protection against NIR radiation, especially because the increase in the glass temperature offers a negligible contribution to the subsequent thermal emission. This opens possibilities for the development of energy-saving smart windows.

## 4. Conclusions

We presented the synthesis of ultrathin silver nanoshells with a high level of extinction in the NIR. The core, made of 200-nanometer silica particles, was first sensitized with Sn^2+^. The addition of Tollens’ reactant resulted in a very dense layer of silver seeds on the silica cores, which enabled ultrathin-shell formation through the reduction of the silver nitrate through consecutive additions of formaldehyde and ammonia. The ultrathin shells, which were either continuous, percolated, or made of isolated silver patches, is 10 nm thick on average. The optical extinction of the particles, associated with high-resolution TEM images, suggests a silver-shell growth mechanism that first starts through the creation of isolated patches of silver on the silica surface. Their subsequent regrowth occurs parallel to the silica surface, leading to a thin continuous silver coating with a remarkably smooth surface. We also showed the effect of different parameters on the synthesis to obtain a reproducible and reliable protocol. Finally, the as-obtained particles were successfully incorporated into transparent thin polymeric films on glass substrates by doctor-blade coating. Particles dispersed in the transparent films decrease NIR transmission; thus, they can be used to prevent solar heating.

## 5. Patents

Marie Plissonneau, Laure Bertry, Thierry Le Mercier, Valérie Buissette, Etienne Duguet, Glenna L. Drisko, Mona Tréguer-Delapierre, Laurent Lermusiaux, Jacques Leng, and Moncef Lehtihet are the inventors and co-applicants of a patent resulting from the work reported in this manuscript (WO2022048833A1) [48].

## Data Availability

Not applicable.

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
