# Peer review of "Silver Nanoshells with Optimized Infrared Optical Response: Synthesis for Thin-Shell Formation, and Optical/Thermal Properties after Embedding in Polymeric Films"

_nanomaterials, 2023, doi:10.3390/nano13030614_

Round 1

Reviewer 1 Report

In this manuscript, “Silver nanoshells with optimized infrared optical response: Synthesis for thin shell formation, and optical/thermal properties after embedding in polymeric films,” the authors present a multi-step, reproducible synthesis of ultrathin silver nanoshells with a strong IR absorption. Based on the obtained results, the authors claimed that the proposed design may be useful for solar control applications. Overall, this manuscript has a strong potential for another review round after applying the issues and addressing the shortcomings listed below:

1-The authors should polish/revise some grammatical mistakes and typos along the manuscript. I invite the authors to read their manuscript carefully and make the required changes where necessary.

2-Please increase size of the text provided in the figures, where necessary.

3-In the Introduction section, while discussing recent developments in the field of nanoparticles, the following works should also be considered and cited to give a more general view to the possible readers of the work: [(i) Plasmonic gadolinium oxide nanomatryoshkas: bifunctional magnetic resonance imaging enhancers for photothermal cancer therapy, PNAS Nexus pgac140 (2022); (ii) Cinnamomum tamala Leaf Extract Stabilized Zinc Oxide Nanoparticles: A Promising Photocatalyst for Methylene Blue Degradation, Nanomaterials 11, 1558 (2021)].

4-There are two “2.2.2” along the manuscript. Please fix this.

5-It seems the authors directly put Scheme 1 from ppt. Please revise Scheme 1 by removing the red underlines and fixing the typos.

6-In Scheme 2, please change the color of Sn2+. It is really hard to read with this color.

7-In Figure 2d, there is no “red line”. Please fix this in the figure caption.

8-In Figure 3, what is the reason for (and origin of) the peak closer to 300nm? Please explain.

Reviewer 2 Report

In this manuscript, the authors proposed a multi-step, reproducible, synthesis of ultrathin silver nanoshells and investigated the shell formation mechanism. However, there are some serious shortcomings for this article, as listed below.

1.       Abstract - please include more actual results rather than general description about the work. Please re-organize.

2.       What is the advantage of ultrathin silver nanoshells? Please clarify

3.       There are a lot of results but little discussion in the Results part. This makes the manuscript like a laboratory report rather than an article. In general, this is very detailed but very superficial, with much written without scientific focus.

4.       The variation in shell thickness is irregular, what is the reason?

5.       Replace Figure 7 as a tablet.

6.        More experiments need to provide to evaluate the optical and thermal properties.

Round 2

Reviewer 1 Report

In its current form, the revised manuscript is suitable for publication.

Author Response

We thank the reviewer for her/his careful review of our manuscript.

Reviewer 2 Report

My comments regarding the previous version of the paper have been taken into account by the authors, who provided satisfactory answers. Consequently, the submitted work can be accepted for publication.

Before it is publised, the authors should proofread and double-check the manuscript very carefully! There are still many typos and improper expressions. For instance,

typo in line 24 page 1 "(poly(vinylpyrrolidone)";

the abbreviations ("SERS" in page 1) used in this book should be expanded at first use;

authors do not explain and mention "Table 1" in the manuscript;

use 'wavelength' instead of '\lambda' in Figure 6 if you use 'wavelength' in previous figures;

authors still use Figure 7 in the revised manuscript since it was removed! 

etc.

Author Response

We thank the reviewer to pick up the inaccuracies in reviewing our manuscript. We have made several changes to the text to clarify it. We feel the manuscript is now significantly more clear. All the changes to the text can be found in yellow in the manuscript attached.